# Thermoplastic Elastomer (TPE)–Poly(Methyl Methacrylate) (PMMA) Hybrid Devices for Active Pumping PDMS-Free Organ-on-a-Chip Systems

**DOI:** 10.3390/bios11050162

**Published:** 2021-05-19

**Authors:** Mathias Busek, Steffen Nøvik, Aleksandra Aizenshtadt, Mikel Amirola-Martinez, Thomas Combriat, Stefan Grünzner, Stefan Krauss

**Affiliations:** 1Hybrid Technology Hub, Institute of Basic Medical Science, University of Oslo, P.O. Box 1110, 0317 Oslo, Norway; steffeno@ifi.uio.no (S.N.); aleksandra.aizenshtadt@medisin.uio.no (A.A.); m.a.martinez@medisin.uio.no (M.A.-M.); thomas.combriat@fys.uio.no (T.C.); stefan.krauss@medisin.uio.no (S.K.); 2Chair of Microsystems, Technische Universität Dresden, 01069 Dresden, Germany; stefan.gruenzner1@tu-dresden.de; 3Department of Informatics, University of Oslo, P.O. Box 1080, 0316 Oslo, Norway; 4Department of Physics, University of Oslo, P.O. Box 1048, 0316 Oslo, Norway; 5Department of Immunology and Transfusion Medicine, Oslo University Hospital, P.O. Box 4950, 0424 Oslo, Norway

**Keywords:** organ-on-a-chip, micro-pneumatics, PDMS-free, thermoplastic elastomers, layer-by-layer manufacturing

## Abstract

Polydimethylsiloxane (PDMS) has been used in microfluidic systems for years, as it can be easily structured and its flexibility makes it easy to integrate actuators including pneumatic pumps. In addition, the good optical properties of the material are well suited for analytical systems. In addition to its positive aspects, PDMS is well known to adsorb small molecules, which limits its usability when it comes to drug testing, e.g., in organ-on-a-chip (OoC) systems. Therefore, alternatives to PDMS are in high demand. In this study, we use thermoplastic elastomer (TPE) films thermally bonded to laser-cut poly(methyl methacrylate) (PMMA) sheets to build up multilayered microfluidic devices with integrated pneumatic micro-pumps. We present a low-cost manufacturing technology based on a conventional CO_2_ laser cutter for structuring, a spin-coating process for TPE film fabrication, and a thermal bonding process using a pneumatic hot-press. UV treatment with an Excimer lamp prior to bonding drastically improves the bonding process. Optimized bonding parameters were characterized by measuring the burst load upon applying pressure and via profilometer-based measurement of channel deformation. Next, flow and long-term stability of the chip layout were measured using microparticle Image Velocimetry (uPIV). Finally, human endothelial cells were seeded in the microchannels to check biocompatibility and flow-directed cell alignment. The presented device is compatible with a real-time live-cell analysis system.

## 1. Introduction

Lab-on-a-chip (LoC) and organ-on-a-chip (OoC) systems are of great interest for biomedical research, pharmaceutical screening, and disease modeling, as well as personalized medicine. Currently, a common and well-established manufacturing technology for LoC and OoC systems is based on soft lithography combined with the two-component elastomer polydimethylsiloxane (PDMS) [1]. In addition to its beneficial material properties such as biocompatibility, high optical grade, oxygen permeability, the possibility to bond PDMS to glass, flexibility, and easy handling characteristics, PDMS also has disadvantages such as its tendency to adsorb small molecules and the need for new master structures for each layout [2]. The adsorption of small molecules into PDMS is especially problematic when it comes to substance testing, as the concentration of the tested drugs is not stable and might be released over time from the PDMS bulk [3]. Furthermore, scalability and variability are issues in PDMS-based platforms. To overcome these issues, a number of new manufacturing technologies for alternative thermoplastic substrates have been developed and established in the last few years. These include hot embossing, laser ablation micro-milling, and 3D printing [4,5,6]. Various materials such as cyclic olefin copolymer (COC), polycarbonate (PC), poly(methyl methacrylate) (PMMA), polystyrene (PS), and polyethylene terephthalate (PET) are widely used [7]. Nevertheless, one major drawback of thermoplastics compared to PDMS is the equipment needed to structure and seal the microdevices [2]. Recent studies have focused on cost-efficient but reliable manufacturing methods such as micro-milled molds and micro-injection molding [8] or laser cutting and stacking of thermoplastic films [9]. The latter still requires expensive laser micro-structuring machines; however, for medium resolution (>100 µm), a cutting plotter [10] and a conventional CO_2_ laser cutter [11] are cost-efficient and versatile alternatives. The latter method is mostly restricted to PMMA as this material can be cut very well with CO_2_ lasers.

With the invention of pneumatically driven microvalves in PDMS-based microfluidic systems by the Quake group [12], fluid manipulation “on-chip” has been enabled. For manufacturing micro-pneumatic systems, a flexible membrane is needed. In OoC systems, pneumatically driven micropumps can be used to stimulate endothelial cells [13] or transport media from one organ model to another [14]. Current multilayer thermoplastic OoCs still integrate silicone films for liquid manipulation [9]. These have to be bonded by adhesion promoters such as APTES, which extends processing times and reduces manufacturing yield [15]. To combine the advantages of thermoplastics and PDMS while avoiding the described disadvantages of both materials, thermoplastic elastomers (TPEs) are a promising alternative [16]. TPEs can be thermally bonded to thermoplastics and easily hot embossed from a master mold. In addition they are flexible, biocompatible, and optically clear, and they have a lower adsorption of small molecules than PDMS [16]. Recent studies have focused on microfluidic fabrication using hot embossing of TPE [16,17,18], as well as on fabrication of thermoplastic–TPE hybrid devices [19,20,21,22]. Micro-pneumatic pumps and valves with a TPE membrane have been successfully developed and are used for perfused well plates [23]. Another main advantage of TPEs is its ability to reversibly bond to different substrates such as glass by heating [24]. This gives the opportunity to extract tissue after the experiment, as well as opens the door for advanced imaging.

A major class of TPEs is represented by multiblock copolymers consisting of soft elastomers and hard thermoplastic blocks, such as styrene block copolymers (SBCs), co-polyesters (COPEs), polyamide/elastomer block copolymers (COPAs), and polyurethane/elastomer block copolymers (TPUs). From all listed material classes, only TPU films are commercially available. For the other TPEs, films have to be produced with a suitable technology. A promising approach is to dissolve the material in a solvent such as toluene, chloroform, or tetrahydrofuran and then use technologies such as spin-coating or casting [25].

Despite their clear advantages, TPEs have a much lower oxygen permeability compared to PDMS, which can have an impact on cell viability. Several studies found oxygen permeability coefficients to be between 2 and 50 Barrer for various TPE materials [26,27]. This is much higher than the oxygen permeability of PMMA (0.15 Barrer [28]) but much lower than that of PDMS (Silpuran^®^ film >500 Barrer [29]).

Here, we present a cost-efficient technology chain to produce flexible films from TPE granulate and to bond these films to laser-cut thermoplastic assemblies. Next, we characterize the bonding quality and the flow characteristics in a TPE-based micropump system. Lastly, we test flow-induced orientation of endothelial cells in a perfusion channel.

## 2. Materials and Methods

A micro-pneumatic system consists of three main components: (1) a pneumatic layer integrating the connections and channels for the pneumatic supply lines, (2) a flexible membrane that moves the fluid and a fluidic part integrating the valves, and (3) microfluidic channels. In this work, we used laser structuring together with thermal diffusion bonding to build up PDMS-free thermoplastic OoCs with an integrated pneumatic micropump. The basic fabrication process is shown in Figure 1. It comprises the following steps:Laser structuring of thermoplastic sheets for the fluidic and pneumatic assembly;Cleaning the parts in an ultrasonic bath;Manufacturing the flexible TPE film (here, spin-coating on an Si wafer);UV treatment of all parts;Thermal bonding in a hot-press;Laser welding of the edges and drilling of connection holes through TPE film.

### 2.1. OoC Material and CO_2_ Laser Machining

PMMA was used as the substrate material for the OoC device, as it has good optical properties, is widely used in cell cultivation due to its biocompatibility, and, importantly, can be processed using versatile and fast CO_2_ laser systems. During laser cutting, PMMA vaporizes into gaseous components, thereby leaving a clean cut. The latter is important for usage in OoC devices, as burned channel walls negatively influence cell viability and optical quality. For the pneumatic part (see Figure 1), a 4/6 mm thick PMMA sheet (RS Components, Corby, UK) was used to allow sufficient depth to connect fluidic and pneumatic connectors. The other parts were made of either PMMA films with a thickness of 250/500 µm (Goodfellow Cambridge Ltd., Huntingdon, UK) or a 1.5 mm thick PMMA sheet (RS Components, Corby, UK).

A 40 W desktop CO_2_ laser cutter was used to structure the substrates (Beambox^®^, Flux Inc., Taipei, Taiwan). Laser ablation was controlled by adjusting the power and the speed of the laser head. Table 1 lists parameter sets for cutting or engraving different thicknesses of PMMA (films/sheets). Engraving depths were measured with a laser triangulation sensor (HG-C1030, Panasonic Europe AG, Holzkirchen, Germany).

### 2.2. TPE Film Production

Different TPE materials have been evaluated in terms of solvent solubility, film manufacture, and handling, identifying the SBC type Styroflex 2G66 (INEOS Styrolution Group GmbH, Frankfurt, Germany) as the best solution for this purpose [30].

We used a solution of 40 wt.-% Styroflex in toluene, mixed it for 24 h until the raw material was completely dissolved, and later spin-coated a film of it. For this, a 3-inch Si wafer was cleaned and baked for 10 min at 120 °C to remove moisture. Next, the solution was poured and distributed at 300 rpm for 30 min (spin-coater: Ossila Ltd., Sheffield, UK). Subsequently, the rotation speed was increased to values between 600 and 900 rpm for 40 s before the film was dried at room temperature for 24 h and finally peeled from the wafer for further processing. Film thicknesses for different rotation speeds (Table 2) were measured with a laser triangulation sensor (HG-C1030, Panasonic Electric Works Europe AG, Holzkirchen, Germany) at 10 different spots on the film.

### 2.3. Thermal Bonding

Both the structured PMMA sheets and the TPE film can be bonded using thermal diffusion bonding. This process is based on the intermolecular diffusion of thermoplastic polymers when heated up to around the glass transition temperature (T_g_). If both surfaces are brought in close contact and pressed together, shorter polymer chains diffuse from one part to the other, thus interconnecting both surfaces. We used a modified pneumatic hot-press (AirPress-0302, Across International LLC, Livingston, NJ, USA) together with an electronic pressure regulator (ITV1030, SMC Corporation, Tokyo, Japan). Bonding pressure was measured with a load cell (TAS606-200 kg, HT Sensor Technology Co Ltd., Xi’an, China). A major drawback of thermal diffusion bonding is that thermoplastics become soft when heated above T_g_ (T_g,PMMA_ = 100 °C), leading to a significant deformation of the channels. To overcome this issue, various methods can be used to locally increase chain motility on the surface of the polymer, e.g., application of solvents (solvent-assisted bonding [31]) or lowering the T_g_ at the surface by irradiation.

Short-wavelength UV light is known to locally break polymer chains at the polymer surface [32]. In this study, an Excimer lamp with 172 nm wavelength and 10 mW/cm^2^ intensity was used (ExciJet172 55-130, Ushio GmbH, Steinhöring, Germany). The lamp was mounted upside down in an open frame (custom-made with laser-cut PMMA sheets). As the gap between the lamp and bottom was fixed to 8 mm, laser-cut PMMA sheets with different thicknesses were placed underneath the substrates in order to adjust the distance between lamp and substrate surface. The lamp was powered by a laboratory power supply, and the exposure time was controlled with a timer relay. UV-assisted bonding was used to sandwich the TPE membrane between the PMMA sheets [19]. 

To characterize the bonding quality, two parameters are of particular interest: the burst load and the deflection of both bonding partners. Bonding strength was measured by applying ascending pressure values with an electronic pressure regulator (ITV1030, SMC Corporation, Tokyo, Japan) until the bond broke. In addition to the bonding strength, the deformation of the parts due to the applied temperature and pressure is an important performance value. A high deformation will lead to clogging of the channel or at least to a negative influence on the flow in microfluidic channels and should be avoided. After the pressure test, delaminated parts were investigated with the profilometer (Dektak 8, Veeco Instruments Inc., Plainview, NY, USA). Moreover, a scanning electron microscope image was obtained with an IT-300 (JEOL Ltd., Tokyo, Japan).

### 2.4. Chip Platform Layout

The setup used in this study consisted of two modules: (A) a microfluidic system containing the micropump, and (B) a perfusion chip for endothelial culture and flow measurements. Each module was in the size of a microscope slide (76 × 26 mm^2^) to fit to standard microscope adapters. The devices were coupled via mini-Luer adapters (Microfluidic ChipShop, Jena, Germany) and silicone tubes (1 mm inner diameter). For a good optical quality, all channels in the cultivation chip were laser-cut (250/500 µm PMMA film) and up- and downstream reservoirs, each with a volume of 200 µL, were integrated.

### 2.5. Pneumatic Pump Driving System

In order to transport fluid in the micro-pneumatic system, the membrane in at least three succeeding chambers has to be pushed down or moved up one after the other, as shown in Figure 2 [33]. This peristaltic motion was produced by a controlling system that was previously developed and characterized by us [33]. It integrates two pneumatic regulators, one for the driving pressure and a second one connected to a Venturi nozzle to produce a defined vacuum. The embedded system now controls up to 24 solenoids to switch between the positive driving pressure and vacuum, thus controlling the actuation state of the pneumatic micro-pumps connected to the controller. Actuation speed and direction can be changed in a graphical user interface for each pump separately.

### 2.6. Flow Measurement with Microparticle Image Velocimetry

The flow measurements were done in the cultivation module (connected to the microfluidic system containing the pump) by means of microparticle Image Velocimetry [34] of 12 µm polystyrene microparticles (Fluoro-Max, ThermoFisher Scientific Inc., Waltham, MA, USA). The experimental setup is comparable to the one used earlier [35] consisting of a standard microscope (Leica MZ APO stereo zoom microscope, Leica Mirosystems, Wetzlar, Germany) coupled to a high-speed camera (acA1300-200uc, Basler AG, Ahrensburg, Germany). Particle movement was observed in a field of view of 592 × 300 Px (1 Px = 5 µm) at 840 fps and 500 µs acquisition time. Later, image stacks of 5000 frames were evaluated using an optical flow algorithm previously applied to calculate the motion of beating cardiomyocytes [36]. The setup is suitable to detect flow velocities in a very broad range (several µm/s up to 1 m/s, depending on the achievable framerate).

### 2.7. On-Chip Endothelial Cell Culture

Human liver-derived endothelial cells (HLECs, Lonza, Basel, Switzerland) and human umbilical vein endothelial cells (HUVECs, Lonza, Basel, Switzerland) were expanded in T75/T175 flasks using endothelial cell growth medium-2 (EGM-2, Lonza, Basel, Switzerland), in a humidified incubator at 37 °C and 5% CO_2_. Before cell plating, the channels in the microfluidic devices were sterilized with 60% ethanol, washed with sterile DI water, and air-dried under sterile conditions. Next, channels were coated with a 0.1% Geltrex solution (Thermofisher Scientific, Waltham, MA, USA) in DMEM/F12 medium for 1 h at 37 °C. HLECs (passage 3–5) and HUVECs (passage 3–5) were detached using Trypsin/EDTA (Sigma-Aldrich, St. Louis, MO, USA) and plated with a concentration of 1 × 10^6^ cells/mL. When the cells had attached and formed a monolayer, (overnight static culture in CO_2_-incubator), the devices containing the cells were coupled to the micropump device and perfusion was started for 24 h.

Perfusion experiments were performed in an IncuCyte (Sartorius AG, Göttingen, Germany) automated live-cell imaging system in the incubator. To check alignment, phase-contrast and fluorescence images were taken every 15 to 60 min.

Cell viability after perfusion was assessed using the Live/Dead Viability/Cytotoxicity Kit (ThermoFisher Scientific, Waltham, MA, USA) according to the manufacturer’s protocol.

After the experiments, endothelial cells in the microfluidic channels were washed with PBS and fixed using 4% paraformaldehyde for 20 min at room temperature (RT). Cells were permeabilized with a 0.1% Triton-X100 solution in PBS (T8787, Sigma-Aldrich, St. Louis, MO, USA) for 15 min at RT and blocked with 5% bovine serum albumin in PBS for 1 h.

Nucleus counterstaining was performed with Hoechst 33342 (Molecular Probes^TM^, ThermoFisher Scientific, Waltham, MA, USA). Staining of actin cytoskeleton was performed using Alexa Fluor 488-phalloidin A12379 (ThermoFisher Scientific, Waltham, MA, USA) at a 1:50 dilution. Stained cells were observed under a standard fluorescence microscope (Zeiss Axio Observer; Zeiss, Oberkochen, Germany); images were acquired using Zen software (Zeiss) and analyzed using Fiji software [37]. Cell counting was performed by adjusting the threshold to a binary image and using the “Analyze Particles” module from Fiji (particles with an area >200 Px^2^ were counted as cells). For quantification of cell orientation, the directionality plugin from Fiji [37,38] with the “Local Gradient orientation method” was used [39].

## 3. Results

### 3.1. Bonding Results—Burst Load and Deflection

In previous studies, the minimum pressure needed to drive a pneumatic micropump (with PDMS membrane) was found to be 50 kPa [33]. Assuming a higher stiffness of the TPE membrane, we set a minimum burst load of 100 kPa for our micro device (safety factor of 2). To investigate the bonding strength, different test devices with a footprint of 20 × 20 mm^2^ and a pneumatic chamber with an area of 200 mm^2^ (round or square) were manufactured. A detailed list of the different assemblies and achieved bonding strengths is given in Appendix B and Table A1.

In general, prior UV treatment allows bonding at temperatures around 80 °C, whereas untreated samples have to be bonded near the glass transition temperature (100 °C). Interestingly, the UV dose needed to induce an effect is only 0.6 J/cm^2^, whereas higher exposure doses might lead to unwanted surface roughing. Very important is the distance between the UV lamp and the substrate, which has to be short since UV light at a wavelength of 172 nm is highly absorbed in air. We found the optimal distance to be between 0.5 and 1 mm. Direct contact to the lamp on the other hand should be avoided because this leads to unwanted substrate heating.

Another major finding was that bonding strength of PMMA films was lower than that of PMMA sheets. This might be due to the fact that films are deflected when pressure is applied, thus peeling them off from the other part. Furthermore, PMMA films are often “impact-modified” to prevent scratches on the surface, especially when used for display covers. This surface hardening might contribute to the lower bonding strength compared to untreated PMMA sheets.

Another reason for delamination is a too low contact area between both bonding partners. For the quadratic chamber, both partners are in contact only at the 2.9 mm wide edges. If the chamber is round, the minimum distance to the edge is only 2 mm. This layout tends to be delaminated at pressure values of only 10–20 kPa due to uneven bonds or local defects. After delamination during the bonding tests, the deflection was measured and a scanning electron microscopy (SEM) image of the delaminated set II was taken (see Appendix A).

In a first run, a 2D scan of the delaminated part was obtained (see Figure 3A). The bottom substrate should be deformed according to the shape of the bonded chamber on top. At the contact areas, the material is compressed (negative deformation). The compressed material then flows into the chamber, thereby leading to a positive deflection (channel collapse). To determine the influence of bonding parameters and material combinations, the deflection was measured in the center of the device and the deflection with respect to the vertical position was plotted. As shown in Figure 3B, a higher bonding pressure led to a significant increase in both the compression at the contact area and the peak deflection in the center of the chamber. At a bonding temperature of 84 °C and a pressure of 2.4 MPa, the maximum compression was 20 µm and the maximum deflection in the middle was only 6 µm. If the bonding pressure was increased to 4.1 MPa, the maximum deflection reached 100 µm and the compression was around 40 µm.

Another finding was that 0.25 and 0.5 mm thick PMMA films were deflected to a higher extent than the 1.5 mm thick PMMA sheet during bonding (see Appendix A). Possible explanations for this can be the shorter polymer chain length and, thus, its lower stiffness, often used for films to reduce the brittleness of PMMA and lateral movement of the film during bonding.

To sum up, the parameters listed in Table 3 for bonding pure PMMA assemblies and PMMA-Styroflex composites were used for chip manufacturing because they led to the lowest deflection while providing sufficient bonding strength.

### 3.2. Microfluidic Layout and Fabrication

As described in the previous paragraph, best bonding results were obtained with the 1.5 mm thick PMMA substrates; thus, the chip was made with several layers of this material. A further conclusion was that the TPE material flows during the bonding process, leading to deformations of the film especially in cavities and at boundaries. This can lead to unwanted bridging or clogging of channels when the parts facing the membrane have different geometries. As a rule, it is advantageous that valves and pumps are designed in a way that deflection of the film does not negatively affect its function, e.g., the use of round chambers that are facing the membrane. Moreover, it has to be considered that a high plastic deformation of the membrane during bonding can have a negative effect on the closing behavior. This can be prevented by constraining the height of the pneumatic chamber to less than 1 mm. A robust and long-term stable pump layout is shown in the exploded view in Figure 4. Based on the actuation cycle shown in Figure 2, all three chambers were designed as valves. The valve chambers were 3 mm in diameter and 500 µm deep, and they featured two holes with a diameter of 500 µm. If pressure is applied, the membrane is pressed into the chamber against these holes and blocks the flow. By applying vacuum, the membrane is deflected backward into the pneumatic chamber, thus filling the valve chamber with liquid from the fluidic layer underneath (laser-engraved channels are 500 µm deep and 1 mm wide). A minor pre-deflection of the TPE membrane does not significantly change the functionality of this valve layout. The stroke of the pump can be adjusted by increasing the diameter of the central pump chamber (diameters of 4 and 4.5 mm were used). The pneumatic part (green) consists of two parts, a connection plate (1) with the pneumatic fittings, tubes, and the engraved pneumatic channels, and a second part (2) integrating the pneumatic chambers. The fluidic part (blue) was built up separately and consists of a plate with structured valve seats (4) and laser-engraved fluidic channels (5). Both the pneumatic and the fluidic parts were first bonded separately, and the spin-coated TPE film was later sandwiched between both assemblies (bonding parameters are listed in Table 3). Lastly, fluidic connections were laser-cut into the TPE film, and the complete device was laser-welded with a power of 28 W and a speed of 3 mm/s by cutting approximately 0.5 mm from each border. This ensured that the edges of the device were stable against delamination.

### 3.3. Flow Measurement

After filling both chips and the tubes with water, PS beads were injected in one of the reservoirs and perfusion was started. The minimum pressure needed for fluid actuation was found to be 60 kPa; below this value, the valves did not close correctly. The driving vacuum for membrane back-movements was set to −80 kPa. The shear stress *τ* for endothelial cells cultivated at the top and bottom of the rectangular channels (height: *h*, width: *w*) can be calculated as follows [13]:(1)$$\tau =\frac{6\mu \cdot Q}{h^{2}\cdot w}.$$

The dynamic viscosity *µ* of water is approximately 1 mPa·s, and the flow rate *Q* can be calculated with the average velocity *v_m_* measured via µPIV.
(2)$$Q=k\cdot v_{m}\cdot w\cdot h.$$

The calibration factor k depends on the flow profile (see Figure A1), the optical setup, and the used flow calculation algorithm, and it was experimentally determined to be 1.25 for a 1 mm wide and 0.25 mm high channel (see channel (1) in Figure 5). In Appendix C, the calibration process is explained. Moreover, we matched the optical flow algorithm with a particle-tracking algorithm on a recorded dataset of particle movement under peristaltic flow and calculated good conformity (average error: 0.17 mm/s, Appendix C). Due to the used valve cycle (Figure 2), a discontinuous flow is generated by the pump [40]. Every time the membranes are deflected, a flow pulse is generated. The peak velocity depends on the stroke volume, the fluidic resistance of the channels, the pressure increase at the membrane, and the volume of the fluidic reservoirs. As the deflected volume and the fluidic resistances are fixed by design, one can only tune the driving pressure of the pump to increase the membrane deflection speed. In the Appendix A, the velocity–time curves for driving pressures of 80 kPa and 150 kPa are compared. The maximum velocity increased from 9 mm/s to 13 mm/s. Interestingly, a higher driving pressure can also be disadvantageous when the valve actuation speed is increased. This effect was reported earlier [20] and can be explained by the high pneumatic dead volume of the tubes that connects the controller and the chip. As the pressurized air needs to be released in order to move the membranes back to filling state, short actuation cycles might lead to an incomplete pressure release and, thus, insufficient membrane movement.

Changing the actuation speed while leaving the driving pressure around 80 kPa was found to be a convenient way to tune the flow velocity and, thus, the wall shear stress. Velocity–time curves for different actuation frequencies (one complete cycle: 0.625/1.2/2.4 Hz) are shown in Figure 6A. As a function these curves, the maximum and mean wall shear stress was calculated using Equation (1) by averaging the flow velocity over one or more complete pump cycles. Figure 6B compares the wall shear stresses for driving pressures of 80 and 150 kPa.

The maximum shear stress at 1.2 Hz was 2.7 dyne/cm^2^ (mean value: 2.1 dyne/cm^2^) and, thus, in the physiological range of endothelial cells (1–20 dyne/cm^2^) [13,41,42]. For frequencies of 0.625 Hz and lower, the flow rate and, thus, the wall shear stress were zero when both out- and inlet valves were switched (phase I in Figure 2). Due to the high volume of the reservoirs up- and downstream of the endothelial channel (see Figure 5), an increase in the switching frequency will lead to a significant offset in the flow rate. This can be explained by liquid from the downstream reservoir being transported to the upstream one, thus leading to a difference in the liquid level in both reservoirs. This difference between the liquid levels in the reservoirs leads to an additional gravity flow component. In this way, the reservoir can be described by a fluidic capacitance that dampens the flow, as previously described by us [43].

### 3.4. Perfused Endothelial Culture

HLECs or HUVECs were introduced into the chamber, and a 24 h perfusion was started after overnight static culture. Initial experiments revealed cell apoptosis and detachment after several hours of pumping. This turned out to be due to small air bubbles transported by the flow (possibly due to leakages at the connectors). Cell damage due to air bubbles could be prevented by increasing the volume of the reservoirs and positioning the perfusion channels lower than the reservoirs causing a de facto bubble trap. This bubble trap design prevented air bubbles from entering the channel. With this design iteration, HLECs and HUVECs remained viable and aligned within 24 h of perfusion. According to live/dead staining, a cell viability of 94.5% ± 3.27% (10 ROIs, 9579 cells) was obtained (a representative image for HUVECs is provided in Appendix A). 

To check if the alignment was flow-induced (and not due to geometric confinement), phase-contrast images of cells at the beginning and after 24 h of perfusion (pump driving pressure ±80 kPa, actuation frequency: 0.3 Hz) were taken and analyzed by Fiji (see Appendix A). Time-lapse videos of the alignment are shown in the Appendix A. At the end of the experiment, cells were fixed and stained on-chip. Figure 7 (left) shows a representative picture of actin filaments (green) and cell nuclei (blue) of HUVECs under perfusion compared to the static control (Figure 7 right). Cells were elongated under flow (longer actin fibers), whereas cells in the static control remained rounder and smaller.

The analysis of the cell orientation revealed a significant effect of the flow on the alignment. In Appendix A, a histogram showing the normalized amount of HLECs oriented relative to flow direction is shown (0°). Histograms for HLECs and HUVECs under flow and a static HUVEC control (each 5–10) were grouped in three orientation categories (0–30°, 30–60°, and 60–90°) and plotted in a whiskers plot (Figure 8).

Under the chosen experimental conditions, endothelial cells responded to shear stress by aligning in the flow direction. A significant difference could be seen in the pre-alignment of the two different observed cell types; at the start of the experiment, 50% of the HLECs were aligned in flow direction (due to the elongated form of the microchannels), whereas HUVECs did not show such a pre-alignment and were oriented in all directions (33.3 %).

Under perfusion (24 h), the number of flow-oriented HLECs increased to 57%. The number of cells oriented from 30 to 60° to the flow direction stayed nearly constant, while the number of cells oriented perpendicular to the flow direction (60–90°) was reduced from 20% to 15%.

With HUVECs, a slight increase in directionality could be observed. The number of cells oriented in the flow direction increased from 36% to 40%, while the number of cells perpendicular to the flow decreased from 32% to 28%. The number of cells oriented between 30 and 60° stayed nearly the same, similar to HLECs. In the static control, no significant change in orientation could be observed.

## 4. Discussion

The aim of this study was to create a PDMS-independent multilayered OoC device with integrated pneumatically driven micropumps. For this, thermoplastic elastomers such as styrene-butadiene block copolymers (SBCs) were found to be suitable alternatives as their hardness is comparable to that of PDMS and some of them are biocompatible and optically clear. Microchannels were cut and engraved in sheets of PMMA using a conventional CO_2_ laser cutter. Later, these sheets were thermally bonded to form pneumatic and fluidic assemblies.

As only a few TPEs are available in the form of films, we had to produce films from SBC. The SBC “Styroflex” was dissolved in toluene and spin-coated on top of a silicon wafer to make films between 50 µm and 100 µm. After drying, these films were bonded and sandwiched between both thermoplastic assemblies. This was done by thermal diffusion bonding. Additional UV treatment prior to bonding was found to increase bonding strength and reduce deformation of both pure PMMA assemblies and PMMA–TPE hybrids. To characterize the bonding performance and carry out pressure testing with suitable parameters, devices were designed and pressure was applied until the devices delaminated. Afterward, the deformation of the demounted parts was measured using a profilometer to check if bonding had a negative effect on the channel geometry.

For pure PMMA bonds, 1 min UV activation at 10 mW/cm^2^ followed by thermal bonding at 84 °C and 2.4 MPa for 15 min was found to be a suitable parameter set. Resulting burst loads were >40 N by a maximum deflection of 6 µm for a 200 mm^2^ large chamber. Without UV activation, the needed bonding temperatures were around 100 °C (the glass transition temperature of PMMA) leading to a high channel deformation of more than 500 µm for the same chamber dimension.

We determined a significant influence of the brand type of the used PMMA material on the bonding strength. This was also reported by others [11] and might be due to differences in the chain length of the polymers or additives and coatings applied during the fabrication process. The bonding strength achievable with our setup is sufficiently high for the fabrication of pneumatic micropumps but is lower than what could be achieved with solvent-assisted bonding (>100 N) [11]. Nevertheless, solvent-assisted bonding tends to be less reproducible and sometimes causes cracks and other defects in the PMMA substrates [44]. Further studies should compare different brands of PMMA substrates, especially with the goal of improving optical quality and solvent resistance.

PMMA–TPE–PMMA hybrids bonded best at 70 °C and 1.6 MPa for 15 min after a UV activation of 1 min at 10 mW/cm^2^. Here, burst loads of more than 100 N could be achieved. Bonding of TPE to PMMA at this temperature led to neglectable deformations of the PMMA substrate but to significant deformations of the TPE films (as they tend to flow under pressure/temperature influence).

A suitable layout for the micropump, especially for the valves, was identified as that shown in Figure 4. In contrast to other designs, only the round pump and valve chambers face the (possibly deflected) membrane so that deformations do not influence the functionality of the device.

The pump-driven flow in a 250 µm high and 1 mm wide channel was evaluated with an adapted PIV setup, and velocity–time curves were in line with previous measurements done on PDMS–thermoplastic hybrid devices with a similar pump layout [13]. Nevertheless, due to the higher flexibility of PDMS compared to our used TPE material, the minimum driving pressure needed to operate the pump increased from 50 kPa to 70 kPa, and peak and mean velocities were slightly higher in the PDMS–thermoplastic hybrid devices. Nevertheless, a direct comparison to other microsystems is difficult as fluidic resistances and driving systems vary, thus resulting in different velocity–time curves. Moreover, the measurements revealed that we can achieve shear stresses in the range of 0.3 to 3 dyne/cm^2^ by tuning the pump pressure or actuation frequency. Long-term stability of the system was proven by repeating the PIV measurements after 7 days of pumping. The PIV measurements did not show significant differences in the velocity–time curves; thus, we could conclude that creep of the TPE did not negatively influence the function. This was also reported by others [22].

Lastly, we could show that HLECs and HUVECs align and elongate with the flow (longer actin filaments) compared to cultivation conditions without flow. Cell viability remained high (95%) in our devices; thus, sufficient oxygen supply can be assumed. This is because the up- and downstream reservoirs are open to the atmosphere, and the cell cultivation medium is so enriched with oxygen. Nevertheless, cell apoptosis could be observed under static conditions, possibly due to the missing convective oxygen transport and poor oxygen permeability of PMMA. In our experiments, therefore, we placed the control on a tilting platform for minimum medium exchange. Our findings are in line with previous reports and can be considered as “medium shear stress stimulation”, where the effect of shear-induced alignment in general is not so pronounced as for shear stresses between 7 and 12 dyne/cm^2^ [13,45]. In future studies, higher shear stresses and prolonged stimulation periods should be investigated and compared to these findings. Higher flow rates could be achievable by increasing the stroke volume or minimizing the pneumatic dead volume. Further studies should also investigate shear effects by applying different flow patterns: continuous, periodic, and aperiodic. Furthermore, it would be beneficial to implement the whole functionality in one microdevice (pump, reservoirs and endothelial channel) to reduce the negative effect of leaking connectors between the pump and cultivation device and to reduce the dead volume of the system.

TPE is a promising replacement material for PDMS due to its lower substance adsorption [23] and good processability [20]. The device design and the absence of external syringe pumps make the system compatible with real-time live cell analysis systems, thereby enabling kinetic studies and overtime monitoring of complex processes such as angiogenesis, migration, and invasion. Taken together, the developed modular perfusion platform is useful for a number of biological questions that are currently difficult to answer otherwise.

We present a cost-efficient fabrication technology of PMMA substrates structured with a conventional CO_2_ laser-cutter, UV-activated with an Excimer lamp, and bonded in a pneumatic hot-press. Table 4 lists the used equipment, suppliers, and prices.

In contrast to other TPE-based micro-pumps [16,20,46], our layout is capable of achieving much higher flow rates (µL/s instead of µL/min), thus also allowing cell stimulation in bigger channels and increased perfusion for cells with higher oxygen consumption rate such as liver tissue. Moreover, our modular approach with a separate pump module connected via tubes to organ-specific OoC devices allows rapid design cycles while reusing the perfusion chip. Furthermore, our micropump produces physiological heart-like perfusion of the organ models. With our layout, tissue modules can be easily pre-cultivated, and different OoC modules can be connected together to form a body-on-a-chip [47]. The CAD files of both the pump chip and the perfusion chip for endothelial cells can be found in the Appendix A.

## Figures and Tables

**Figure 1 biosensors-11-00162-f001:**
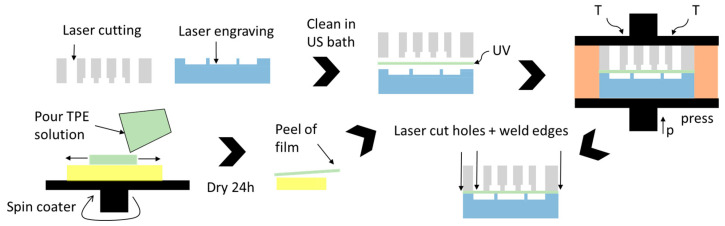
Chip fabrication process: laser structuring of a pneumatic (gray) and a fluidic part (blue). TPE film production (green) on a Si wafer (yellow). UV treatment and thermal bonding.

**Figure 2 biosensors-11-00162-f002:**
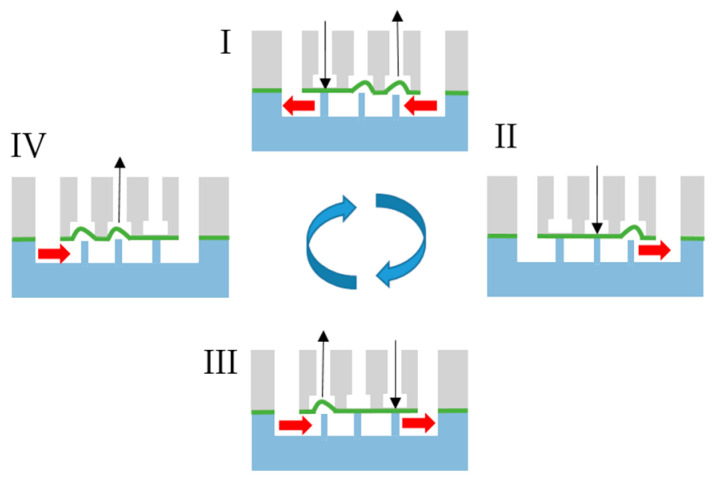
Four-step actuation cycle generated by the pneumatic controlling unit.

**Figure 3 biosensors-11-00162-f003:**
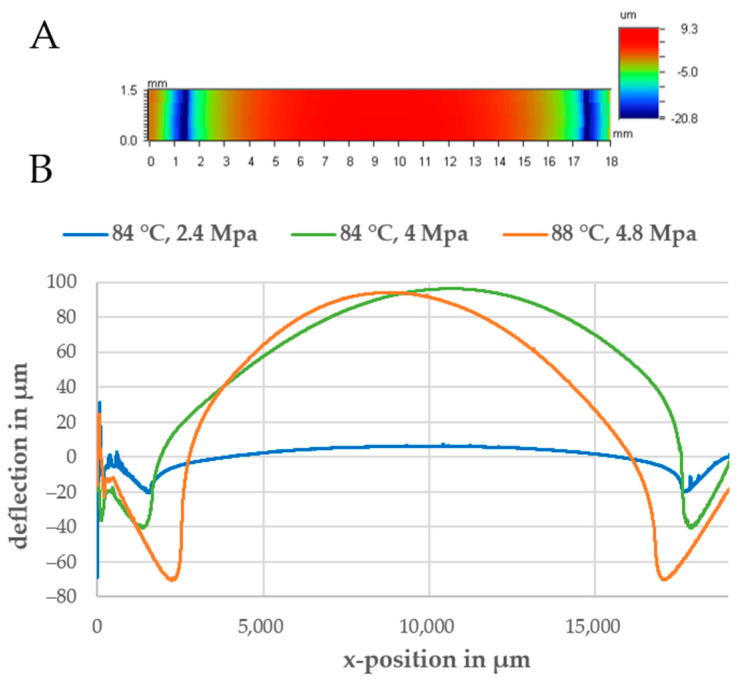
(**A**) A 2D scan of the bottom part after delamination to a round pneumatic chamber. (**B**) Line scans of the bottom part for different bonding parameters.

**Figure 4 biosensors-11-00162-f004:**
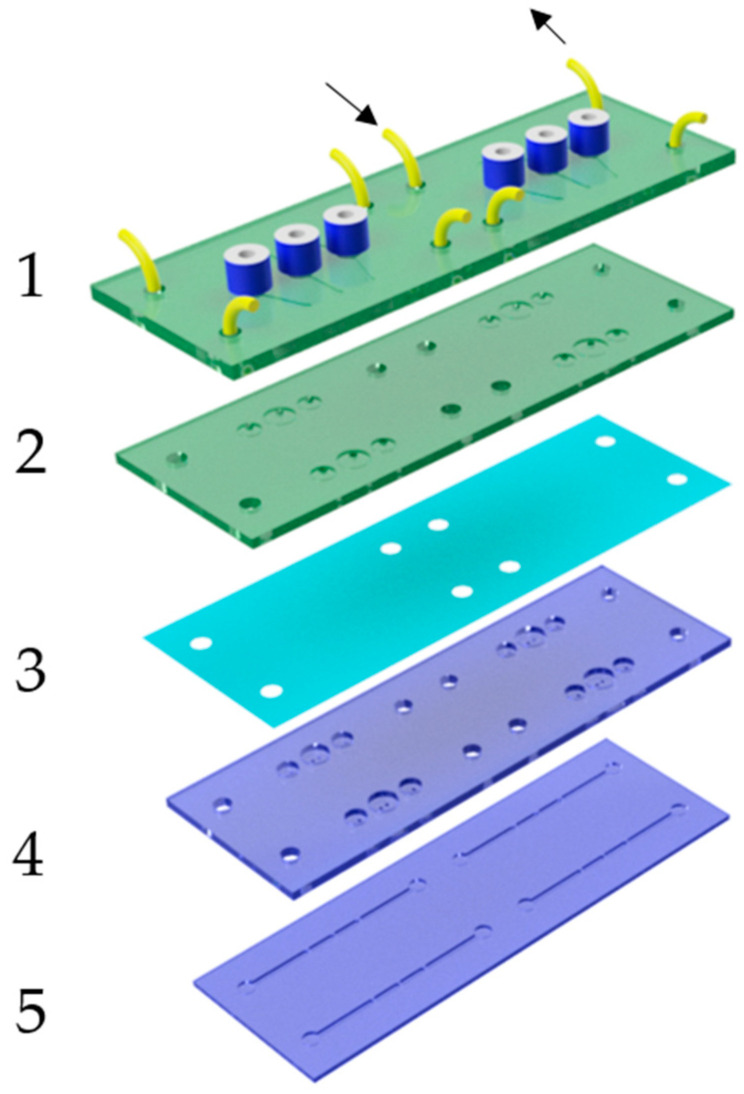
Exploded view of the micropump. Green: pneumatic layers (1) and (2) with connectors. (3) TPE membrane. Blue: fluidic assembly with valve seats (4) and fluidic channels (5).

**Figure 5 biosensors-11-00162-f005:**
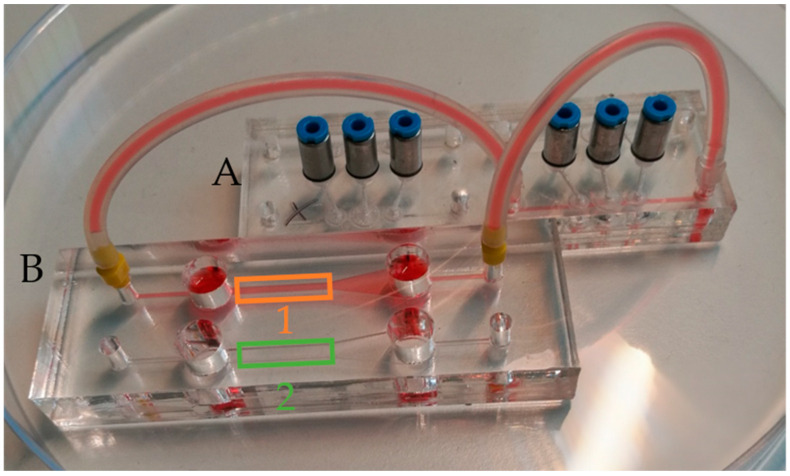
Cultivation chip (**B**) coupled to the micropump chip (**A**) using tubes. The two-perfusion channels are 1 mm (1) and 2 mm (2) wide and 0.25 mm high. Reservoirs are 6 mm in diameter and 6 mm deep.

**Figure 6 biosensors-11-00162-f006:**
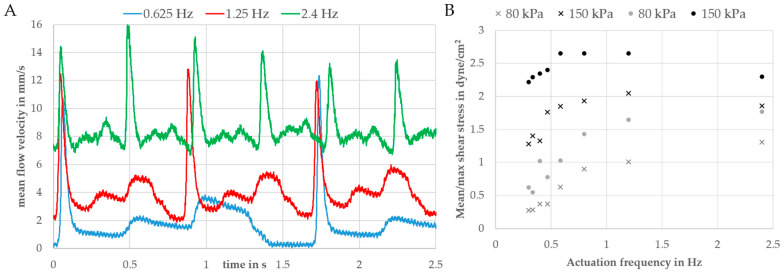
(**A**) Velocity–time curves for three different actuation frequencies and a driving pressure of ±70 kPa. (**B**) Maximum (dots) and mean (crosses) wall shear stresses in a 1 mm wide and 0.25 mm high channel. Calculated using Equation (1) for different driving pressures and actuation frequencies (negative driving pressure: −80 kPa).

**Figure 7 biosensors-11-00162-f007:**
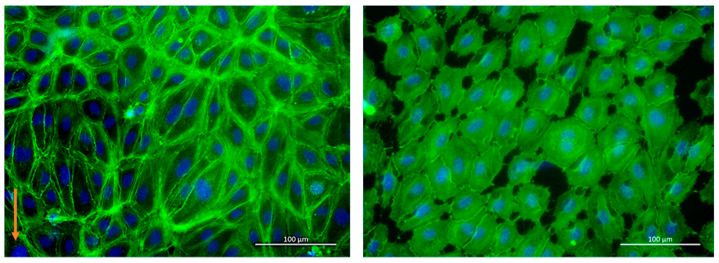
Stained HUVECs with (**left**) and without (**right**) perfusion. Green: actin filament, blue: nuclei. The orange arrow indicates the flow direction.

**Figure 8 biosensors-11-00162-f008:**
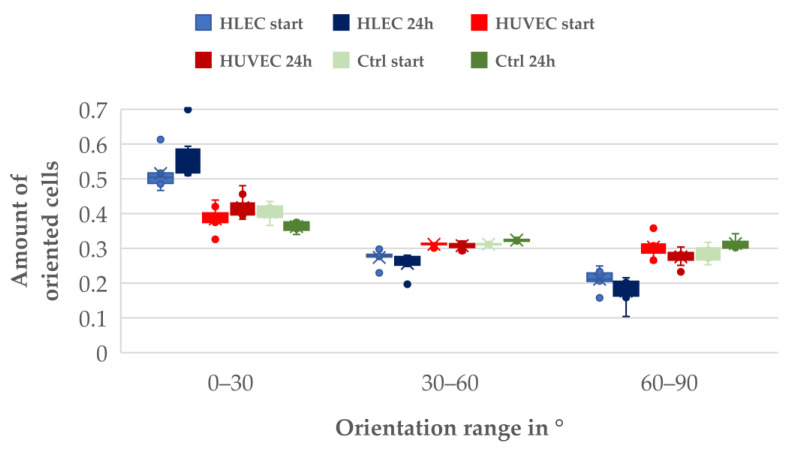
Orientation with respect to flow direction of different endothelial cells (6–10 frame pairs analyzed for each dataset).

**Table 1 biosensors-11-00162-t001:** Power/speed setting for cutting (right) and engraving (left) of PMMA films and sheets.

	Engraving			Cutting					
**Depth in mm**	0.25	0.5	1.5	4	0.25	0.5	1.5	4	6
**Power in W**	4.8	6	8	12	6	8	12	24	24
**Speed in mm/s**	25	40	40	25	15	20	15	8	4

**Table 2 biosensors-11-00162-t002:** TPE film thicknesses (after drying) at different rotation speeds (40 wt.% Styroflex 2G66).

**Spin Speed in rpm**	500	600	700	800	900
**Thickness in µm**	72 ± 6	67.5 ± 8.6	48.3 ± 13	43 ± 8.9	38.2 ± 10.8

**Table 3 biosensors-11-00162-t003:** Optimal bonding parameters for pure PMMA and PMMA–TPE assemblies.

	UV Dose in J/cm^2^	Bonding Pressure in MPa	Bonding Temperature in °C	Bonding Time in min
PMMA–PMMA	0.6	2.4	84	15
PMMA–TPE	0.6	1.6	70	15

**Table 4 biosensors-11-00162-t004:** List of used equipment, prices and suppliers.

	Brand	Supplier	Price
CO_2_ laser cutter	Beambox, 40 W	https://www.fluxlasers.com/beambox.html (accessed on 7 May 2021)	3000 EUR
Pneumatic hot press	AirPress-0302	https://www.acrossinternational.com/ (accessed on 7 May 2021)	600 EUR
Excimer lamp	ExciJet172 55-130	https://www.ushio.co.jp/en/products/1002.html (accessed on 7 May 2021)	5000 EUR
Spin Coater	Ossila	https://www.ossila.com (accessed on 7 May 2021)	2200 EUR

## Data Availability

Data is contained within the article or Appendix A.

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
