# Peer review of "Thermoplastic Elastomer (TPE)–Poly(Methyl Methacrylate) (PMMA) Hybrid Devices for Active Pumping PDMS-Free Organ-on-a-Chip Systems"

_biosensors, 2021, doi:10.3390/bios11050162_

Round 1
Reviewer 1 Report
The manuscript " Thermoplastic elastomer (TPE) – Poly-(methyl methacrylate) (PMMA) hybrid devices for active pumping PDMS-free Organ-on-a-Chip systems " by Busek et al. presents a new method to fabricate TPE films as membranes for microfluidic applications. Replacing PDMS with other thermoplastic materials has been proposed as a strategy to address the shortcoming of PDMS-based microfluidic devices. Overall, this is a valuable and practical idea that might reduce further complications of using PDMS in microfluidic devices. The paper is well-structured and contains relevant and fairly new information. However, a few more experiments need to be conducted, or more data should be added to the main manuscript to improve the quality of the manuscript. Here are more detailed comments:
- Introduction: The introduction has a good structure and stated the limitation and disadvantages of PDMS, but it is somehow short. The history of the most recent advancement of thermoplastic-based microfluidic devices has not been comprehensively included. It would be useful to describe where these devices can be used or what the progress is in the field, especially for organ-on-a-chip platforms. If there are some works in the literature, what is the difference between the current work from those?
- Results: Page 5, where the threshold for pressure is defined, it would be better if the authors explain why 100 kPa is the threshold and how this number varies for different applications.
- Results: It was thoroughly described the procedure for using a UV source to initiate bonding. It would be helpful if the authors suggest how the optimal distance of 0.5 and 1 mm was achieved as this might be challenging for others to do using regular laboratory tools.
- Results: Figure 3 should have been mentioned and explained before Figure 4. Or It is wrongly named as figure 4a (Page 6, line 241)
- Results: in line 253, it is not clear that 0.25/0.5 means 0.25 and 0.5 or anything else? Please clarify this.
- Results: Line 274, "Therefore, the chambers should be less than 1 mm high" probably tall should be used instead of high! There are a few more grammar issues in the paper that need to be addressed.
- Results: Please mention the physiological range of shear stress for endothelial cells in line 341.
- Results: Figure 7 was wrongly labeled as figure 3 on page 10. The same happened for Figure 8 on page 11.
- Results: it would be useful to perform burst pressure for TPE membranes with different thicknesses.
- Results and discussion: the accuracy of flows at different frequencies should be mentioned and compared with other works.
- Results: Cell viability assays such as live and dead assay and MTT needs to be included, and the microscopic images that show the orientation of the cells should be added to the main manuscript (it is hard to see the orientation of the cell from the images that are included in the supplementary). Probably longer perfusion is needed to see the impact of flow.
Reviewer 2 Report
The following comments may help the authors to revise the manuscript before acceptance.
- The authors should also address the reasons for choosing the current materials for their fabrication. There are other materials such as COC, COP, PS, and other fabrication techniques, for examples, micro-milling and injection moulding. References can be used: https://www.mdpi.com/2072-666X/10/9/624.
- Name and manufacturer of the CO2 laser machine?
- The process in figure 1, should it be done in a cleanroom or a typical lab environment?
- The authors should list the names of the equipment and price so that it can be helpful for the readers to acquire.
Reviewer 3 Report
This paper demonstrates TPE-PMMA hybrid microfluidic devices that can be used for organ-on-a-chip applications. It is an important study that uses only plastic-based materials which is highly relevant to traditional cell-culture environments. The experimental design of endothelial cell culture also is pertinent as a showcase of these hybrid devices. It is well-written and systematically designed the experiments. I added a couple of comments to improve the quality of the manuscript. I would recommend to accept this paper in the journal of Biosensor after revisions.
The authors discussed the low oxygen permeability of hard plastic in Introduction, what about the oxygen permeability of TPE? Are there any references?
Line 36: type error Polydimethylsiloxane: no capital letter “P”
Line 56: Recent study also shows TPE-plastic thermally-bonded devices without applied pressures. (Moon et al., Processes, 9 (1), 54, 2021)
Line 137: There is also a recent paper that uses solvent-assisted bonding (AMD Wan et al., Lab Chip. 2015;15(18):3785-92)
Line 173, Figure 2: The flow direction is not clear. Is the fluid flows left to right?
Line 334 Figure 6A: Why the initial point of the flow velocity is different? The 0.625 Hz starting level seems to be higher than 1.2 and 2.5 Hz.
Line 363-364: “For quantification, the ….. was used” this sentence can be moved to the Materials and Methods section.
Line 368: Figure 3 should be Figure 7.
Line 383: Figure 4 should be Figure 8.
Apart from the endothelial cell align and elongation of the flow direction, it is also important to show/discuss cell viability of TPE-based approach compared to PDMS or PMMA.
Reference 33: Is this a company name?
Round 2
Reviewer 1 Report
I appreciate the authors' effort to address raised comments. The paper has been significantly improved and can be accepted in the current format.
Reviewer 2 Report
I do not have other comments.